# A Decade of Antifungal Leads from Natural Products: 2010–2019

**DOI:** 10.3390/ph12040182

**Published:** 2019-12-12

**Authors:** Mohammed Aldholmi, Pascal Marchand, Isabelle Ourliac-Garnier, Patrice Le Pape, A. Ganesan

**Affiliations:** 1Department of Natural Products and Alternative Medicine, College of Clinical Pharmacy, Imam Abdulrahman Bin Faisal University, Dammam 31441, Saudi Arabia; mjaldholami@iau.edu.sa; 2School of Pharmacy, University of East Anglia, Norwich Research Park, Norwich NR4 7TJ, UK; 3Université de Nantes, Cibles et Médicaments des Infections et du Cancer, IICiMed, EA 1155, F-44000 Nantes, France; pascal.marchand@univ-nantes.fr (P.M.); isabelle.ourliac@univ-nantes.fr (I.O.-G.); patrice.le-pape@univ-nantes.fr (P.L.P.)

**Keywords:** fungal pathogens, antifungal agents, natural products

## Abstract

In this review, we discuss novel natural products discovered within the last decade that are reported to have antifungal activity against pathogenic species. Nearly a hundred natural products were identified that originate from bacteria, algae, fungi, sponges, and plants. Fungi were the most prolific source of antifungal compounds discovered during the period of review. The structural diversity of these antifungal leads encompasses all the major classes of natural products including polyketides, shikimate metabolites, terpenoids, alkaloids, and peptides.

## 1. Introduction

The global increase in antimicrobial resistance among pathogenic bacteria, viruses, fungi, and parasites is a serious concern for human healthcare. In the case of fungi, more than one billion individuals worldwide are affected by fungal infections and the associated mortality, over 1.5 million deaths each year, is equivalent to that caused by tuberculosis and more than triple that of malaria [1]. Although relatively rare in healthy individuals, the incidence of superficial and invasive fungal infections has dramatically risen in recent years. This is due to a growing ‘at-risk’ population with impairments in their immune system, breaches in physical barriers to fungal entry, or an altered microbiome. Skin mycoses are predominantly caused by *Trichophyton*, *Microsporum,* and *Epidermophyton* genera while *Candida*, *Cryptococcus*, *Aspergillus,* and *Pneumocystis* genera, and *Mucorales* are the most common invasive fungal pathogens [2]. Meanwhile, emerging pathogenic fungi that are either new species such as the recently described *Candida auris* [3] or well-known species spreading in their ecological distribution represent additional threats to human health.

The growing challenges posed by fungal diseases are further heightened as antifungal treatment is mainly limited to the azoles and echinocandins. The azoles are the most widely used antifungals and are synthetic compounds that reversibly inhibit cytochrome P450-dependent lanosterol or eburicol 14α-demethylase with moderate specificity for the fungal enzyme over the human counterpart [4]. Nevertheless, they suffer from off-target toxicity as well as issues with fungistatic rather than fungicidal activity in yeasts that promote the development of resistance. The major resistance mechanisms to azoles involve genetic mutations or increased expression of the target enzyme, or amplification or induction of efflux pumps. The echinocandins are fungal lipopeptide natural products (Figure 1) that are non-competitive inhibitors of 1,3-β-glucan synthase, an enzyme involved in fungal cell wall biosynthesis. While the natural products are not optimal in terms of pharmacokinetics, three semisynthetic derivatives are approved for clinical use: anidulafungin prepared from echinocandin B, caspofungin prepared from pneumocandin B_o_, and micafungin prepared from FR901379 [5]. Although the selectivity of the echinocandin target for fungi provides a good safety profile, these compounds are large peptides, requiring intravenous administration, while mutations at hotspots in the target enzyme lead to resistance. In addition to the azoles and echinocandins, the polyenes and pyrimidines are two other classes approved for antifungal therapy. The natural product polyenes (Figure 2) are macrolides isolated from various *Streptomyces* strains. The prototypical amphotericin B has been in clinical use for the treatment of systemic fungal infections since the 1950s and is still an important option in critical cases. Several additional polyenes, nystatin, natamycin, hamycin, and filipin, have received regulatory approval. As a class, the polyenes have significant nephrotoxicity due to their relatively nonselective mechanisms of ergosterol binding and pore formation within the cell membrane [6,7]. Finally, synthetic pyrimidine antimetabolites such as flucytosine interfere with nucleic acid biosynthesis, but resistance via point mutations in the fungal uracil phosphoribosyltransferase or cytosine deaminase enzymes restricts their application to combination therapy [8].

Overall, the current drugs have numerous limitations including toxicity, drug–drug interactions, poor pharmacokinetics, narrow spectrum of activity, and fungistatic versus fungicidal action. These inherent liabilities are exacerbated in immunocompromised patients since their immune system cannot effectively assist in the eradication of the infection, thus requiring complex and prolonged treatment regimens [9]. A further alarming trend is the rising incidence of fungal clinical isolates that are resistant to the currently used antifungals [10,11]. The scale of the problem is highlighted by the fact that the newest class of approved antifungals, the echinocandins, were actually discovered fifty years ago. The American Food and Drug Administration (FDA) has recognized the need for new antifungals by placing *Candida* and *Aspergillus* on their list of qualifying pathogens [12]. Therapies directed against these species will benefit from incentives including an additional five-year marketing exclusivity besides eligibility for designation as a fast-track drug.

## 2. A Pipeline of Antifungal Natural Product Leads

While antifungal agents with novel mechanisms of action are in various stages of clinical development, their number is relatively small compared to other therapeutic indications [13]. A pipeline of additional preclinical leads is clearly needed, and natural product screening is an important contributor in this regard. One unique feature of natural products is their high structural diversity, sampling areas of chemical space that are difficult to access through purely synthetic compounds [14,15]. Natural products are also well validated to possess biological activity, with many examples approved as therapeutic agents either in their native form or as semisynthetic derivatives [16]. For this review, we searched the online database *Natural Product Updates* for publications that reported novel natural products with antifungal activity from January 2010 to November 2019. From the publications, we selected novel natural products that were active against human pathogenic fungi with an MIC < 10 μg/mL or IC_50_ < 10 μM. In the discussion, we include any information on additional biological activity observed or mechanistic studies on the mode of action. The compounds are classified below according to the type of producing organism.

### 2.1. Natural Product Antifungal Leads from Bacteria and Algae

Actinomycetes are the most prolific source of bacterial natural products, and this remains the case for recently discovered antifungal leads (Figure 3, Figure 4, Figure 5 and Figure 6, **1–29**). In addition, there were three examples isolated from non-actinomycete species (Figure 7, **30–35**) and two from algae (Figure 8, **36–37**). A strain of *Streptomyces albolongus* YIM 101047 isolated from elephant dung produced a number of bafilomycins in laboratory fermentation. The new example 21-deoxybafilomycin A1 (**1**) and the sesquiterpene (1β,4β,4aβ,8aα)-4,8a-dimethyloctahydronaphthalene-1,4a(2*H*)-diol (**2**) displayed antifungal activity against *Candida parapsilosis* with an MIC of 3.2 μg/mL, while being inactive against other species [17]. Genome sequencing of the strain suggested the presence of forty-six putative biosynthetic gene clusters [18]. In the course of biosynthetic labelling experiments, it was discovered that supplementation by acetate produced new metabolites in a *Streptomyces hyaluromycini* MB-PO13 strain. Among these, rubromycin CA1 (**3**) was active against Gram-positive bacteria and *Candida albicans* NBRC 1594 with an MIC of 6.3 μg/mL, whereas an analogue with an additional alcohol was inactive [19]. A strain of *Actinoalloteichus* isolated from marine sediment was the source of neomaclafungins A–I (**4–12**), a series of macrolides of the oligomycin family of antibiotics. The neomaclafungins were active against *Trichophyton mentagrophytes* with MIC values between 1 and 3 μg/mL, compared to 10 μg/mL for oligomycin A [20].

Fermentation of a *Streptomyces* sp. isolated from mangrove rhizosphere soil led to the isolation of a series of azalomycin F natural products (**13–20**) with MIC values of 1.6–6.3 μg/mL against *C. albicans* as well as antibacterial and cytotoxic activity [21,22]. Astolides A (**21**) and B (**22**) are polyol macrolides isolated from *Streptomyces hygroscopicus* collected from alkaline soil [23]. The compounds have MICs of 1–2 μg/mL against *C. albicans*, *Candida tropicalis,* and *Aspergillus niger*. Related macrolides caniferolides A–D (**23–26**) were isolated from the marine-derived *Streptomyces caniferus* CA-271066 [24]. Like the astolides, the caniferolides displayed potent antifungal activity with MICs of 0.5–2 μg/mL against *C. albicans* and 2–8 μg/mL against *Aspergillus fumigatus*, as well as similar levels of cytotoxicity against human tumor cell lines. Caniferolide A was also shown to have in vitro activity against targets relevant to Alzheimer’s disease [25]. Enduspeptides A–C (**27–29**) are depsipeptides that differ in the acyl chain attached to the threonine residue and were isolated from a *Streptomyces* sp. The peptides had an IC_50_ of 2–8 μg/mL against *Candida glabrata* [26].

Within the period under review, three antifungal leads were isolated from non-actinomycete bacterial strains. Fermentation of a myxobacterial *Nannocyctis* sp. led to the isolation of nannocystin A (**30**) with a novel macrocyclic scaffold. While the compound inhibited *C. albicans* with an MIC_50_ of 73 nM, it also inhibited human cancer cell lines at a nanomolar level [27]. The mechanism of action involves binding to the eukaryotic translation elongation factor 1α and structure–activity relationships have been established through the total synthesis of analogues [28]. The burkholdines are lipopeptide antifungal agents previously isolated from *Burkholderia ambifaria* 2.2N, with three new examples Bk-1119, Bk-1213, and Bk-1215 (**31–33**) displaying potent activity against *C. albicans* and *A. niger* [29]. Among the burkholdines, Bk-1119 was the most active against *A. niger* with an MIC of 0.1 μg/mL and also had the best antifungal/hemolytic ratio. Additional analogues were prepared by total synthesis [30]. The Gram-negative bacterium *Chitinophaga pinensis* DSM 28390 produces the novel lantibiotics pinensins A and B (**34**, **35**). Although lantibiotics are typically antibacterial, the pinensins were only weakly so while having MICs of 2–4 μg/mL against yeasts and filamentous fungi [31].

The marine alga *Laurencia* is a prolific producer of secondary metabolites. The sesquiterpene eudesma-4(15),7-diene-5,11-diol (**36**) isolated from a Red Sea sample of *Laurencia obtusa* was antifungal with MIC values of 2–7 μM against *Candida* and *Aspergillus* species [32]. The prenylated xylene caulerprenylol B (**37**) was isolated from the green alga *Caulerpa racemosa* and had MIC_80_ values of 4 μg/mL against *C. glabrata* and *Cryptococcus neoformans* while being inactive against *A. fumigatus* [33].

### 2.2. Natural Product Antifungal Leads from Sponges

Marine sponges are an important source of novel natural products, and more than ten examples with antifungal activity were described in the above-mentioned period (Figure 9 and Figure 10, **38–55**). Extracts from the symbiotic two-sponge association *Plakortis halichondroides*−*Xestospongia deweerdtae* yielded a number of peroxide natural products, of which plakinic acids I, J, K, and L (**38–41**) were potent against *Candida* and *Cryptococcus* species with MIC ≤ 0.5 μg/mL [34]. Plakinic acid M (**42**) was active against *Cryptococcus gattii*, *Cryptococcus grubii,* and *Candida krusei* with MIC_90_ values of 2.4–3.4 μg/mL but less active against *C. albicans* [35]. Extraction from the South China Sea sponge *Hippospongia lachne* was the source for hippolachnin A (**43**), a polyketide with an unprecedented scaffold [36]. The compound was potently antifungal with an MIC of 0.4 μg/mL against *C. neoformans*, *Trichophyton rubrum,* and *Microsporum gypseum*. However, the natural product and analogues obtained by total synthesis were inactive, suggesting the initial report was in error [37]. Bioassay-guided fractionation of the same extract led to isolation of a racemic sesterterpene hippolide J (**44**) [38]. The natural product was resolved into its two enantiomers, and both were highly potent antifungals with MIC_50_ values of 0.13–0.25 μg/mL against *Candida* and *Trichophyton* while weakly cytotoxic to the human embryonic kidney HEK293 cell line.

A new member of the manzamine alkaloids, zamamidine D (**45**), was isolated from an Okinawan marine sponge *Amphimedon* sp. Zamamidine D had an IC_50_ of 2 μg/mL against *C. neoformans* but was weakly active against other fungal and bacterial strains tested [39]. From another Okinawan marine sponge *Pseudoceratina* sp., ceratinadin A and B (**46**, **47**) were isolated with MIC values of 4 and 8 μg/mL, respectively, against *C. neoformans* and 2 and 4 μg/mL, respectively, against *C. albicans* [40]. From an extract of the sponge *Pseudaxinella reticulata*, several crambescin guanidine containing alkaloids were isolated. Crambescin A2 392 and 406 (**48**, **49**) inhibited *C. neoformans* with MIC_50_ values of 1.2 and 0.9 μg/mL, respectively, while being relatively inactive against *C. albicans* [41]. The enantiomers of two known crambescins, crambescin A2 420 (**50**) and Sch 575948 (**51**), were also isolated with MIC_50_ values of 1.1 and 2.5 μg/mL, respectively, against *C. neoformans.* Among metabolites isolated from the marine sponge *Agelas*, two new diterpene alkaloids from *Agelas citrina*, agelasidine E and F (**52**, **53**), were reported to have MIC values of 8 and 4 μg/mL, respectively, against *C. albicans* [42]. Isoagelasine C (**54**), isolated from *Agelas nakamurai*, had an MIC value of 4.7 μg/mL against *C. albicans* [43]. Ageloxime B (**55**), isolated from *Agelas mauritiana*, had an IC_50_ value of 5.0 μg/mL against *C. neoformans* as well as antibacterial activity [44].

### 2.3. Natural Product Antifungal Leads from Plants

Plants accounted for nearly ten antifungal leads within the last decade (Figure 11 and Figure 12, **56–64**). The flavonoid (*E*)-6-(2-carboxyethenyl) apigenin (**56**) was isolated from an extract of *Mimosa caesalpiniifolia* Benth., a Brazilian medicinal plant commonly known as “sabiá” or “sansão-do-campo” [45]. The compound inhibits *C. krusei* with an IC_50_ of 44 nM, although it was inactive against *C. glabrata*. The isoflavonoid vatacarpan (**57**) with an MIC of 1 μg/mL against *C. albicans* was isolated by bioassay-guided fractionation from the roots of *Vatairea macrocarpa* (Benth.) Ducke [46]. The biaryl ether laevicarpin (**58**) was isolated from leaves of *Piper laevicarpu*, known as “falsa-pimenteira” in Brazil [47]. Interestingly, the compound was previously prepared synthetically prior to this isolation. Laevicarpin had an IC_50_ of 7.9 μM against *C. gattii*, in addition to an IC_50_ of 50 μM against the trypomastigote form of *Trypanosoma cruzi*. The dimeric chalcone kamalachalcone E (**59**) was isolated from the red dye extracted from whole uncrushed fruits of *Mallotus philippinensis* [48]. The chalcone exhibited an IC_50_ of 4–8 μg/mL against two strains of *C. neoformans*.

Investigation of the juvenile leaves of *Eucalyptus maideni* F. Muell led to the discovery of a number of phloroglucinol derivatives, among which eucalmaidial A (**60**) showed antifungal activity against *C. glabrata* with an IC_50_ of 0.8 μg/mL [49]. A monoterpene indole alkaloid, 16,17-epoxyisositsirikine (**61**), isolated from the evergreen shrub *Rhazya stricta* Decne. had an IC_50_ of 6.3 μg/mL against *C. glabrata* but was less active against other *Candida* species tested [50]. Erchinine B (**62**), a monoterpene indole alkaloid with an unusual embedded 1,4-diazepine ring, was isolated from roots of *Ervatamia chinensis* and had an MIC of 6.3 μg/mL against *T. rubrum*, with a lower MIC of 0.8 μg/mL against the Gram-positive bacteria *Bacillus subtilis* [51]. An aporphine alkaloid (**63**) was isolated from the bark of a Costa Rican sample of *Beilschmiedia alloiophylla* [52]. The alkaloid had an MIC of 8 μg/mL against *C. albicans*, as well as antileishmanial activity and inhibition of acetylcholinesterase. The cyclic peptide tunicyclin D (**64**) was isolated from roots of the medicinal herb *Psammosilene tunicoides* W. C. Wu et C. Y. Wu [53]. The peptide exhibited MIC_80_ values of 0.3–16 μg/mL against *Candida* species and 1.0 μg/mL against *C. neoformans*.

### 2.4. Natural Product Antifungal Leads from Fungi

Within the last decade, fungi were the most prolific source of novel antifungal leads (Figure 13, Figure 14, Figure 15, Figure 16 and Figure 17, **65–98**). An extract of the endophytic species *Pestalotiopsis mangiferae* obtained from the leaves of the plant *Mangifera indica* Linn. yielded an unprecedented epoxyacetal 4-(2,4,7-trioxa-bicyclo[4 .1.0]heptan-3-yl) phenol (**65**) with an MIC of 0.04 μg/mL against *C. albicans* strains and 1.3 μg/mL against the bacterium *Micrococcus luteus* [54]. Two phenalenones, auxarthrone A and D (**66**, **67**) were obtained from fermentation extracts of an *Auxarthron pseudauxarthron* strain isolated from rabbit dung [55]. The compounds have MIC values of 3.2 and 6.4 μg/mL, respectively, against *C. neoformans* and *C. albicans*. Further investigation into these compounds demonstrated that they are unnatural artifacts, arising from the reaction of natural products with ketone solvents employed during the extraction. Grifolaone A (**68**) was isolated from the edible mushroom *Grifola frondosa*. Interestingly, the hemiketal lactone was obtained in an optically active form and assigned as the *S* enantiomer [56]. The furanone was a potent inhibitor, MIC of 0.15 μg/mL, of the opportunistic human pathogen *Pseudallescheria boydii* and also had an MIC of 10 μg/mL against *A. fumigatus*.

The tropolone nemanolone B (**69**) was isolated from fermentation of a *Nemania* sp. fungus and displayed antifungal activity with an IC_50_ of 4.5 μg/mL against *C. albicans*, and similar levels of activity against the parasite *Plasmodium falciparum* and human tumor cell lines [57]. The quinone pleosporallin E (**70**), isolated from a marine-derived *Pleosporales* sp., inhibited *C. albicans* with an MIC of 7.4 μg/mL [58]. Five new isocoumarins were isolated from fermentation of an endophytic *Pestalotiopsis* sp. obtained from *Photinia frasery*. Among these, pestalactone C (**71**) inhibited *C. glabrata* with an MIC_50_ value of 3.5 μg/mL [59]. Aspergillusether D (**72**), isolated from fermentation of *Aspergillus unguis* PSU-RSPG204, inhibited *C. neoformans* with an MIC value of 8 μg/mL, and inhibited *C. albicans* at a lower level [60]. A series of *p*-terphenyl natural products was isolated from a strain of *Floricola striata* inhabiting the lichen *Umbilicaria* sp., among which the quinones floricolin B and C (**73**, **74**) displayed MIC_80_ values of 8 μg/mL against *C. albicans* [61]. Further investigation of floricolin C suggested a fungicidal action through disruption of mitochondria [62].

Extended fermentation (365 days) of a marine-derived strain of *Aioliomyces pyridodomos* led to the appearance of new metabolites, of which onydecalin C (**75**) had an MIC of 2 μg/mL against *Histoplasma capsulatum* [63]. The same strain, in a more conventional fermentation period (25 days), produced aintennol A (**76**) with an IC_50_ of 8 μg/mL against *H. capsulatum* [64]. Genome mining for potential Diels–Alderase enzymes identified a potential candidate in the genome sequence of *Penicillium variabile*. The putative biosynthetic gene cluster was engineered into an *Aspergillus nidulans* expression host, enabling the isolation of varicidin A (**77**) with an MIC_50_ value of 8 μg/mL against *C. albicans* [65]. The *N*-demethylated analogue, varicidin B, was two-fold less active. In the same manner, the ilicicolin H biosynthetic gene cluster including a putative Diels–Alderase from a producing strain*, Neonectria* sp. DH2, was heterologously expressed in *A. nidulans*. In addition to ilicicolin H, a shunt metabolite ilicicolin J (**78**) was isolated with an MIC of 6.3 µg/mL against *C. albicans* [66]. Heterologous expression was also employed to confirm the biosynthetic gene cluster involved in the production of the burnettramic acids A and B (**79** and **80**) in *Aspergillus burnettii* FRR 5400 [67]. Burnettramic acid A had an MIC value < 1 μg/mL against *C. albicans* while burnettramic acid B was slightly less active with values of 1–2 μg/mL.

Coculture of two extremophilic fungal strains of *Penicillium fuscum* (Sopp) Raper & Thom and *Penicillium camembertii/clavigerum* Thom isolated from a single sample of surface water from Berkeley Pit Lake led to the production of novel metabolites. Berkeleylactone A (**81**) displayed modest antifungal activity with an IC_50_ of 6 μg/mL against *C. glabrata* and higher antibacterial activity [68]. Fermentation of a Saudi strain of *Petriella setifera* led to the identification of the triterpene glycoside amnomopin (**82**) with MIC values of 0.5–2 μg/mL against *Candida* species [69]. Sclerodol B (**83**), a triterpene from extracts of the endophyte *Scleroderma* UFSM Sc1(Persoon) Fries obtained from *Eucalyptus grandis,* had an MIC of 6.3 μg/mL against *C. krusei* with weaker activity against other species [70]. A strain of the marine-derived fungus *Stachybotrys chartarum* produced several novel diterpenoids, of which atranone Q (**84**) had an MIC of 8 μg/mL against *C. albicans* and weaker antibacterial activity [71].

An endophytic *Penicillium* sp. isolated from grass produced picolinic acid derivatives in fermentation. Penicolinate B and C (**85**, **86**) had MIC values of 1.5 and 3.7 μg/mL, respectively, against *C. albicans* [72]. The didymellamide series of pyridone alkaloids was isolated from cultures of the marine-derived fungi *Stagonosporopsis cucurbitacearum* and *Coniochaeta cephalothecoides* [73,74]. Didymellamide A, F, and G (**87–89**) were antifungal with MIC values of 3 μg/mL against *Candida* species. The fermentation also yielded (+)-*N*-hydroxyapiosporamide (**90**), the enantiomer of the previously isolated natural product, with an MIC value of 6.3 μg/mL against *C. albicans*. Fermentation of a *Cyathus* cf. *striatus* basidiomycete led to the isolation of the alkaloid pyristriatin A (**91**) with an MIC of 8.3 μg/mL against *Rhodotorula glutinis* and similar levels of activity against Gram-positive bacteria and human tumor cell lines [75].

The alkalophilic extremophilic fungus *Emericellopsis alkalina* VKPM F-1428 was the source of the peptaibol emericellipsin A (**92**), which exhibited antifungal MIC values of 2–4 μg/mL against *Candida* and *Aspergillus* species as well as activity against Gram-positive bacteria. Bioassay-guided fractionation of extracts of *Colispora cavincola* isolated from plant litter led to the discovery of the linear peptides cavinafungin A and B (**93**, **94**) [76]. The cavinafungins inhibited *Candida* species with an MIC of 0.5−4 μg/mL and *A. fumigatus* at 8 μg/mL. However, the antifungal effects were lost in the presence of mouse serum. Cavinafungin A also potently inhibits the Zika and dengue virus, with the mechanism of action attributed to inhibition of the host signal pepdidase [77]. The antifungal activity of *Phaeosphaeria* sp. F-167,953 was ascribed to the lipodepsipeptide phaeofungin (**95**) with some structural similarity to the previously known phomafungin [78]. Phaeofungin had an MIC of 4 μg/mL against *Trichophyton mentagrophytes* and lower activity against other fungi tested.

High-throughput screening by Astellas Pharmaceuticals against a silkworm model of *A. fumigatus* infection led to bioassay-guided fractionation activity of an extract of *Acremonium persicinum* MF-347833. The siderophore hexapeptide ASP2397 (**96**) was discovered as an aluminum chelate with exceptional potency against *A. fumigatus*, with an MIC of 0.2 μg/mL and efficacy at 3.2 mg/kg in a mouse in vivo model [79]. The metal-free form AS2488059 (**97**) as well as the congener AS2524371 (**98**) were also isolated, and the target was identified as a fungal siderophore transporter [80,81]. The compound was out-licensed to Vical and renamed VL-2397, reaching Phase II clinical trials that were recently discontinued. From another strain of *A. persicinum*, similar cyclic peptides acremonpeptides A–D were isolated in which the asparagine residue is replaced by serine, alanine, phenylalanine, or tryptophan [82]. Surprisingly, these compounds were inactive in antifungal or antibacterial assays at concentrations up to 100 μg/mL, suggesting that the asparagine residue is important for antimicrobial activity.

## 3. Discussion

Between 2010 and 2019, our literature survey identified nearly a hundred novel natural products reported with promising antifungal activity against human pathogens. The compounds originate from a variety of organisms comprising bacteria, algae, fungi, sponges, and plants. The distribution between these sources (Figure 18) indicates that fungi and bacteria were the most common source of antifungal compounds. Indeed, this follows historical trends where the approved antifungals of natural product origin arise from either fungi (the echinocandins) or actinomycetes (the polyenes). The techniques employed in the publications under review range from classical phytochemical studies with plants to high-throughput screening of extract collections and modern microbiological strategies such as cocultivation and heterologous expression of biosynthetic gene clusters. All the major classes of natural products including polyketides, shikimate metabolites, terpenoids, alkaloids, and peptides are represented. As the majority of examples in this review involve the initial disclosure of activity, further investigations are needed to assess the therapeutic potential of highly active compounds as well as their selectivity as antifungal agents. Such studies will need to take into account the ability to produce larger quantities of the natural product. While this is often possible by scaling up the original isolation protocol, additional options are available that involve chemical total synthesis, engineered production in the native strains, or using heterologous hosts. These strategies open up the possibility of the discovery of ‘unnatural’ analogues that may be superior in their pharmacodynamic or pharmacokinetic properties compared to the original natural product.

It is interesting to observe the physicochemical space occupied by these natural product leads (Table 1). Although the compounds are diverse in their structural features, they are largely compliant with the typical guidelines for small molecule drug-like chemical space. While many of the natural products are large in molecular weight, resulting in an average of 569, other properties such as hydrogen bonding potential, molecular flexibility, and polarity often remain within the recommended limits.

## Figures and Tables

**Figure 1 pharmaceuticals-12-00182-f001:**
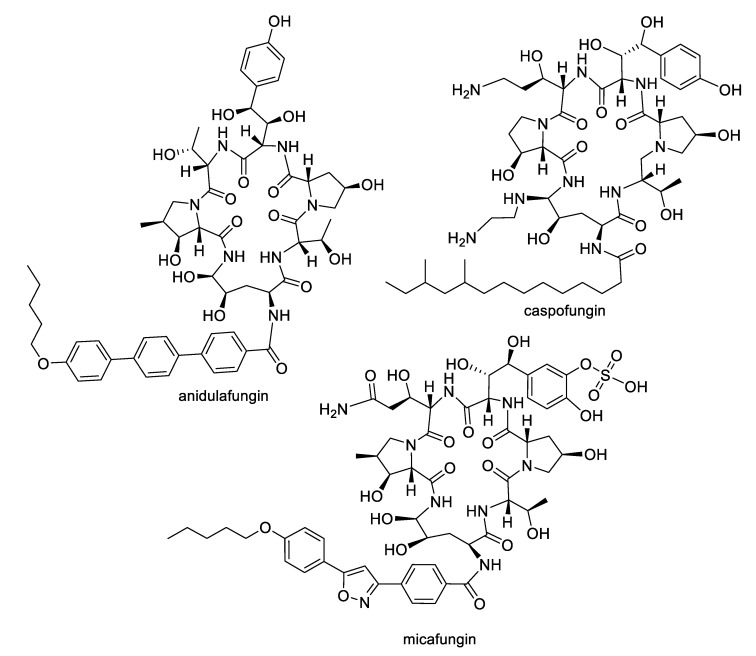
Semisynthetic derivatives of the echinocandin family of natural products approved for antifungal therapy.

**Figure 2 pharmaceuticals-12-00182-f002:**
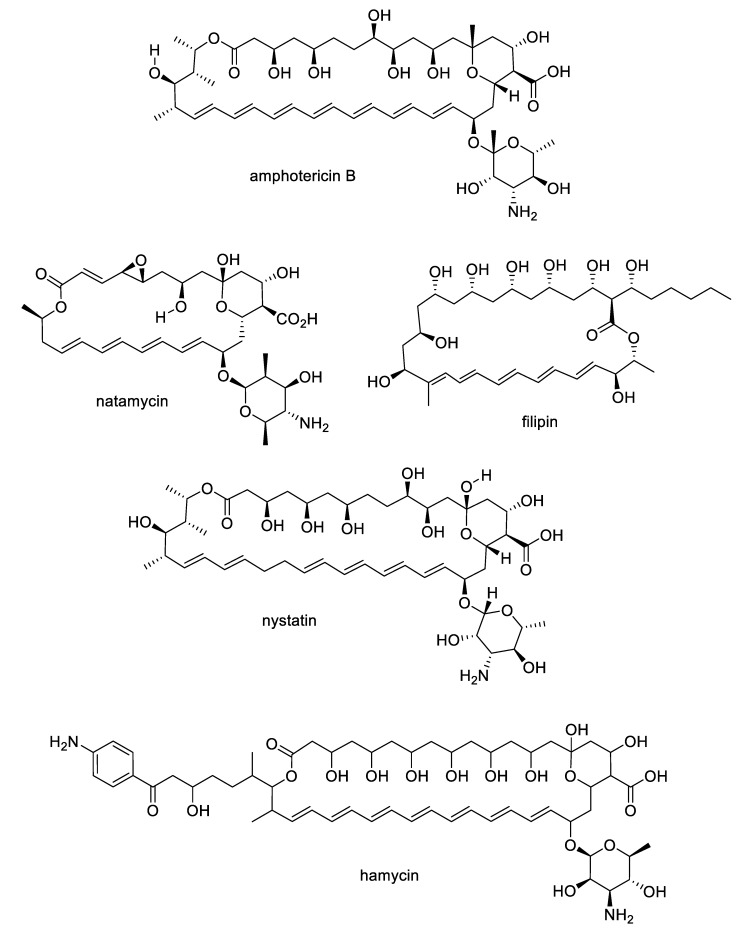
Polyene natural products approved for antifungal therapy.

**Figure 3 pharmaceuticals-12-00182-f003:**
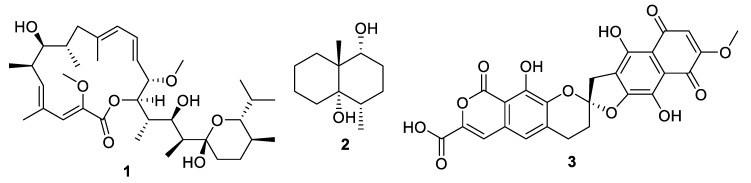
Structures of natural products **1**–**3**.

**Figure 4 pharmaceuticals-12-00182-f004:**
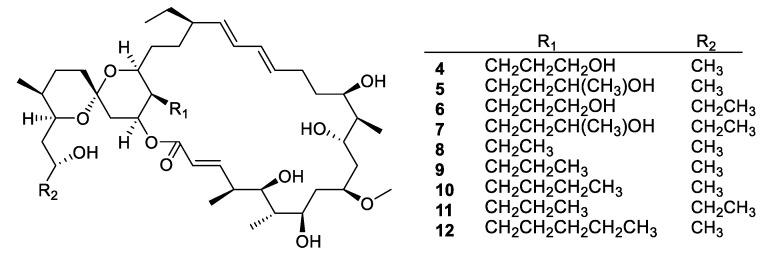
Structures of neomaclafungins A–I (**4**–**12**).

**Figure 5 pharmaceuticals-12-00182-f005:**
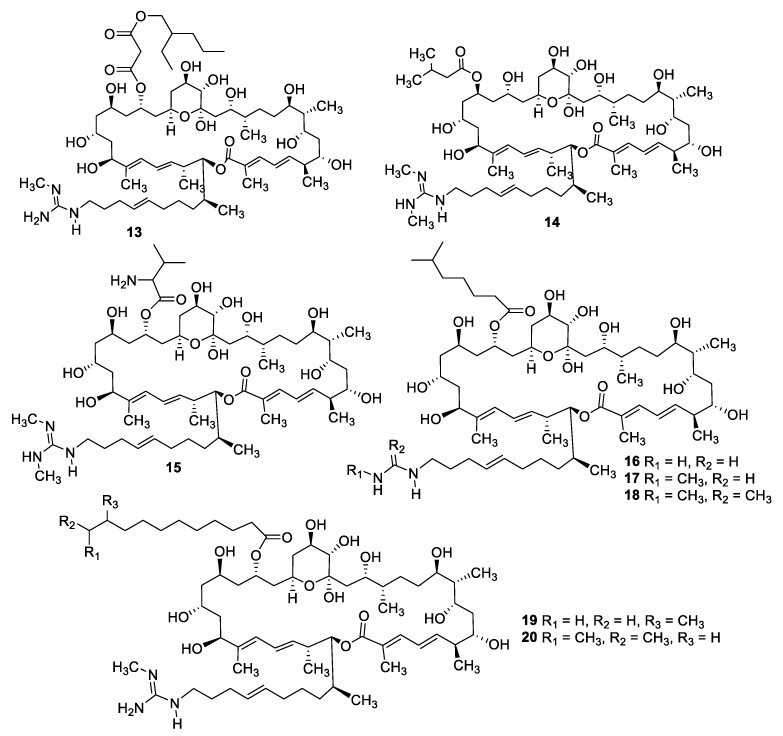
Structures of azalomycin F macrolides **13**–**20**.

**Figure 6 pharmaceuticals-12-00182-f006:**
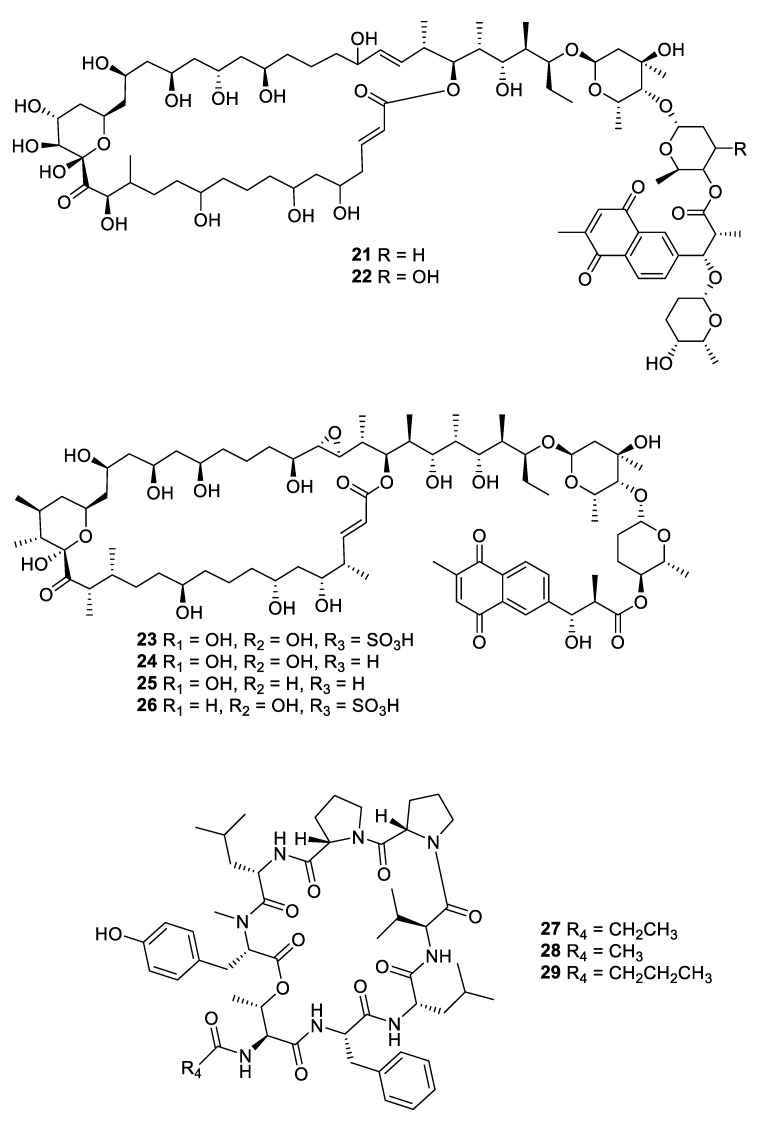
Structures of macrolide and depsipeptide natural products **21**–**29**.

**Figure 7 pharmaceuticals-12-00182-f007:**
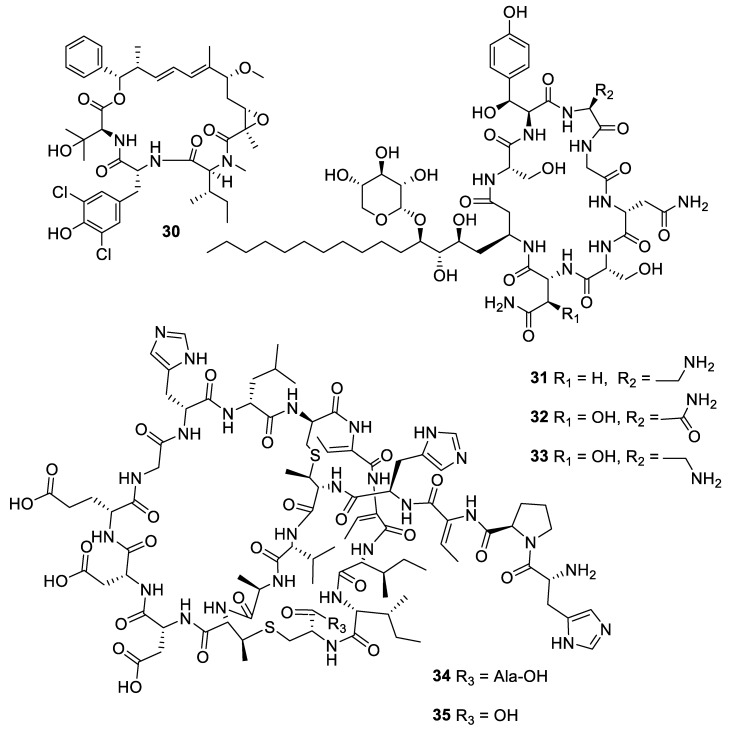
Structures of macrocyclic bacterial natural products **30**–**35**.

**Figure 8 pharmaceuticals-12-00182-f008:**
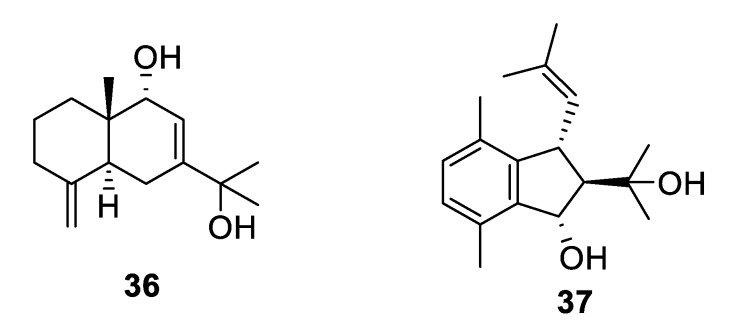
Structures of natural products **36** and **37** obtained from algae.

**Figure 9 pharmaceuticals-12-00182-f009:**
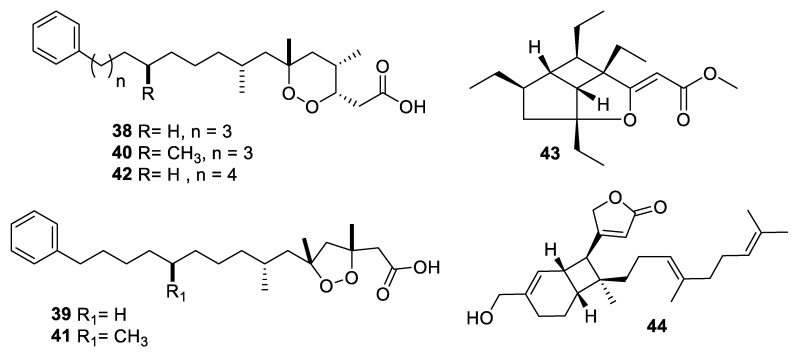
Structures of natural products **38**–**44** isolated from sponges.

**Figure 10 pharmaceuticals-12-00182-f010:**
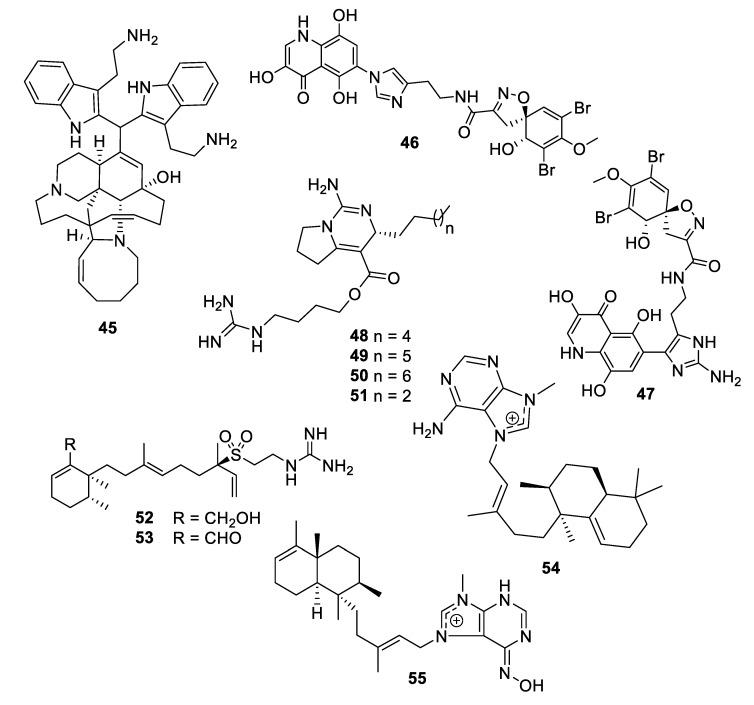
Structures of natural product alkaloids **45**–**55**.

**Figure 11 pharmaceuticals-12-00182-f011:**
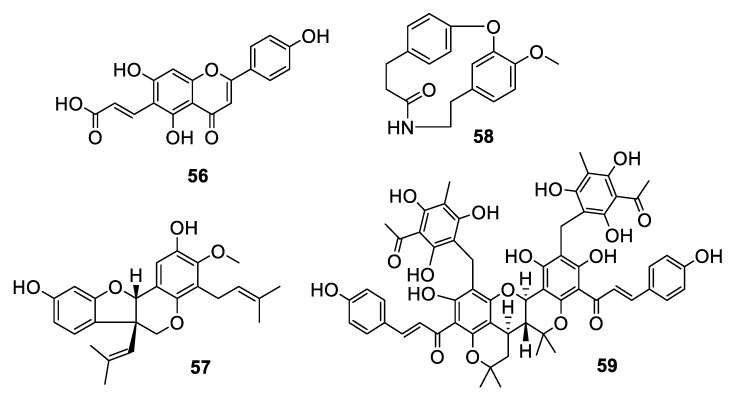
Structures of aromatic natural products **56**–**59**.

**Figure 12 pharmaceuticals-12-00182-f012:**
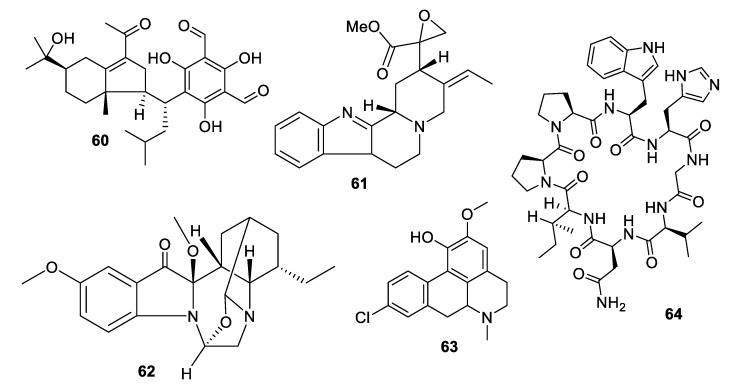
Structures of natural products **60**–**64** isolated from plants.

**Figure 13 pharmaceuticals-12-00182-f013:**
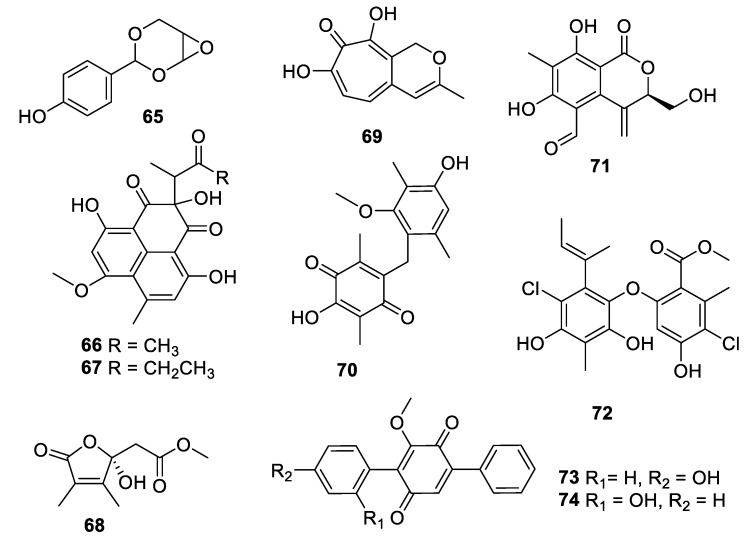
Structures of natural products **65**–**74** isolated from fungi.

**Figure 14 pharmaceuticals-12-00182-f014:**
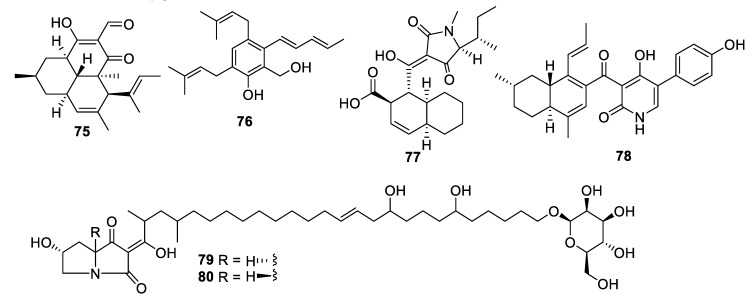
Structures of natural products **75**–**80** isolated from fungi.

**Figure 15 pharmaceuticals-12-00182-f015:**
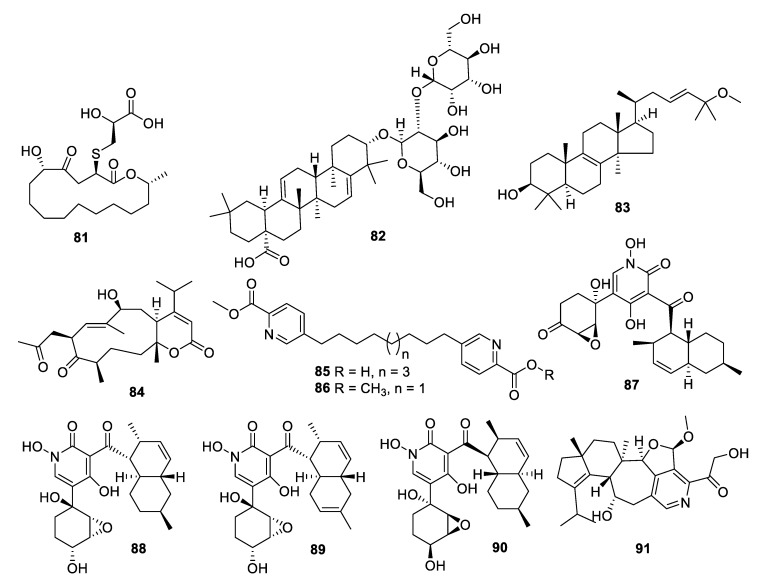
Structures of natural products **81**–**91** isolated from fungi.

**Figure 16 pharmaceuticals-12-00182-f016:**
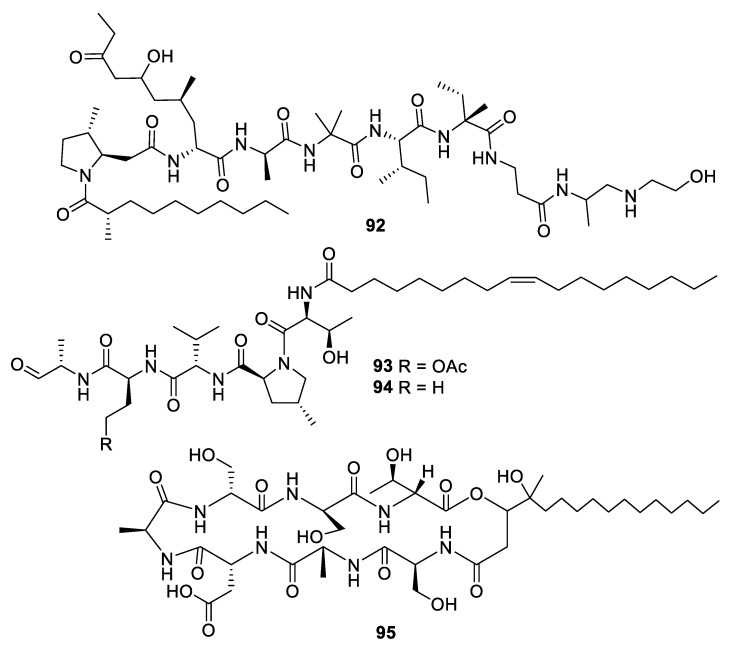
Structures of peptide natural products **92**–**95**.

**Figure 17 pharmaceuticals-12-00182-f017:**
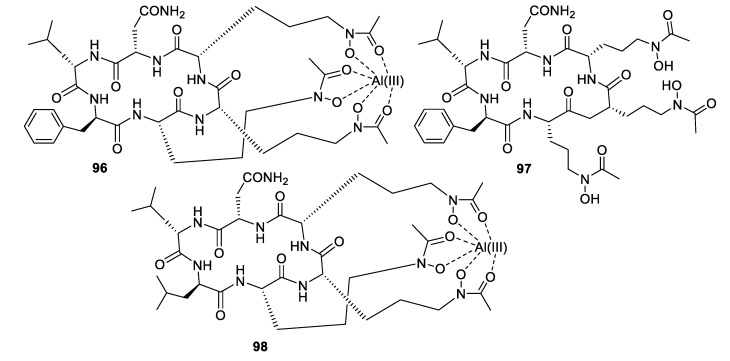
Structures of siderophore natural products **96**–**98**.

**Figure 18 pharmaceuticals-12-00182-f018:**
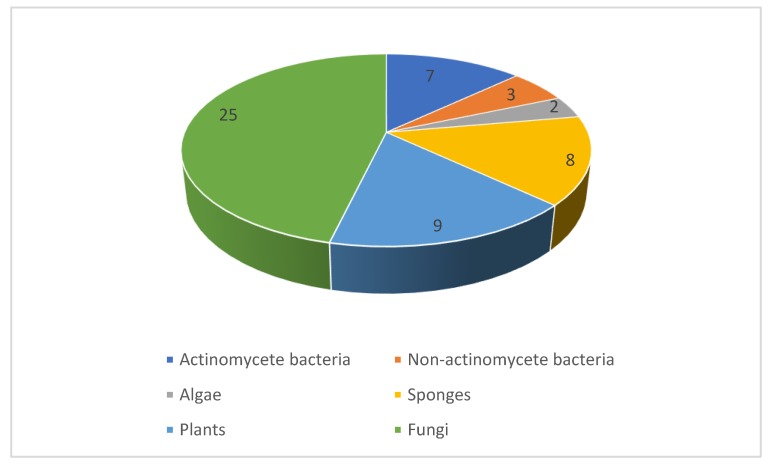
Source of the antifungal natural products discussed in this review. The numbers indicate the number of unique scaffolds from each source.

**Table 1 pharmaceuticals-12-00182-t001:** Physicochemical properties of antifungal natural products. MW = molecular weight, clogP = calculated log P, HBD = hydrogen bond donors, HBA = hydrogen bond acceptors, nrot = number of rotated bonds, TPSA = total polar surface area in Å^2^. The values were taken from SciFinder (https://scifinder-n.cas.org), based on calculations using Advanced Chemistry Development (ACD/Labs) Software V11.02. In certain cases where the data was absent in SciFinder, values were calculated using the Molinspiration website (https://www.molinspiration.com/). For natural products for which a series of related compounds was reported, one representative example was selected. Shaded cells indicate values above the recommended guidelines for small molecule drug-like chemical space (MW ≤ 500, Clog *p* ≤ 5, HBD ≤ 5, HBA ≤ 10, nrot ≤ 10, TPSA ≤ 140).

Compound	MW	clogP	HBD	HBA	nrot	TPSA
**1**	607	4.8	3	8	10	115
**2**	198	2.1	2	2	2	41
**3**	508	2.9	4	12	5	186
**8**	751	7.0	5	10	10	155
**17**	1123	7.6	13	18	26	312
**21**	1580	0.9	15	29	33	472
**28**	987	4.7	6	19	11	253
**30**	817	4.3	4	12	9	167
**31**	1200	−5.7	23	32	36	546
**35**	2144	−0.4	26	55	30	876
**36**	236	2.4	2	2	3	41
**37**	274	3.0	2	2	4	41
**42**	419	7.7	1	4	14	56
**44**	385	6.2	1	3	9	47
**45**	713	8.1	7	7	10	110
**46**	667	0.7	6	13	10	188
**48**	393	3.1	6	8	13	130
**53**	438	5.0	4	6	12	121
**54**	423	2.0	2	5	5	61
**55**	439	2.3	2	6	5	70
**56**	340	1.9	4	7	6	124
**57**	423	6.0	2	5	7	68
**58**	297	2.7	1	4	1	48
**59**	1065	8.0	11	18	23	319
**60**	487	6.5	4	7	12	132
**61**	352	3.0	0	5	3	64
**62**	370	0.5	0	6	3	51
**63**	281	2.5	1	3	2	33
**64**	901	−0.1	10	21	9	303
**65**	194	1.3	1	4	2	51
**66**	358	3.0	3	7	6	121
**68**	200	−1.6	1	5	4	73
**69**	206	−1.0	2	4	2	67
**70**	316	3.5	2	5	5	84
**71**	264	0.8	3	6	5	104
**72**	427	7.9	3	6	8	96
**74**	306	4.7	1	4	4	64
**75**	329	6.8	1	3	3	54
**76**	327	6.7	2	2	9	41
**77**	376	3.3	2	6	5	95
**78**	432	5.4	3	5	4	87
**79**	770	3.2	8	13	35	218
**81**	405	2.3	3	7	6	146
**82**	779	6.4	8	13	14	216
**83**	457	9.5	1	2	6	30
**84**	391	2.7	1	5	3	81
**85**	399	5.1	1	6	14	89
**87**	444	1.4	3	8	6	128
**90**	446	3.5	4	8	7	131
**91**	442	3.5	2	6	6	89
**92**	1064	4.9	10	20	38	294
**93**	792	6.6	5	14	31	200
**95**	904	−2.0	13	23	23	368
**97**	891	−2.5	11	23	21	339
**Average**	569	3.4	5	10	11	155

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
