# Peer review of "A Decade of Antifungal Leads from Natural Products: 2010–2019"

_pharmaceuticals, 2019, doi:10.3390/ph12040182_

Round 1

Reviewer 1 Report

The manuscript describes natural products and related compounds with antifungal activity reported during 2010-2019. Overall, it is well organized and easy for the readers to understand. For the better quality publication, I would like to request for additional information regarding the trends of a decade of antifungal leads compared to before 2010. The authors insist that “fungi were the most prolific source of novel antifungal leads” on pages 1 and 10, however, it is not clear what you based on that evaluation on, that may be an increase in the number of compounds or publications, or structural diversity. It can be helpful to add a chart that shows the numbers of each category, bacteria, alga, fungi, etc., to discuss the most prolific source trends in antifungal leads of the decade. In addition, it is meaningful to discuss the background why “fungi were the most prolific source” during 2010-2019, such as technical development, etc.

Minor mistake: In Figure 9 on page 8, “1” should be “n”.

Author Response

The manuscript describes natural products and related compounds with antifungal activity reported during 2010-2019. Overall, it is well organized and easy for the readers to understand. For the better quality publication, I would like to request for additional information regarding the trends of a decade of antifungal leads compared to before 2010. The authors insist that “fungi were the most prolific source of novel antifungal leads” on pages 1 and 10, however, it is not clear what you based on that evaluation on, that may be an increase in the number of compounds or publications, or structural diversity. It can be helpful to add a chart that shows the numbers of each category, bacteria, alga, fungi, etc., to discuss the most prolific source trends in antifungal leads of the decade. In addition, it is meaningful to discuss the background why “fungi were the most prolific source” during 2010-2019, such as technical development, etc.

Minor mistake: In Figure 9 on page 8, “1” should be “n”.

We added a new Figure 19 which displays the source of antifungal leads according to the type of organism as a pie-chart.

The Figure 9 is corrected.

Reviewer 2 Report

General impression:

The manuscript by Aldholmi and co-workers is a very nice piece of work reflecting extensive literature search performed by the authors. The manuscript is adequately structured and can be easily followed. I have no major criticism about the content of the manuscript and thus strongly endorse publication of this manuscript as review article in Pharmaceuticals. Before that, I would kindly ask the authors to incorporate some minor modifications in the manuscript as suggested below:

Minor comments:

In the entire manuscript: species names should be given as full names (e.g. Candida albicans) when mentioned for the first time and then the abbreviated genus name should be used (C. albicans). This is convention in the microbiology research community and should also be followed in this manuscript.

Figure 2: For consistency, the sugar moieties should all be presented as chair presentations (as in natamycin) or as stereo projections (as in the other ones).

line 61: “In summary” sounds like a conclusion, however, this is only the introduction section. If possible, rephrase this sentence to have a good connecting passage to the main section.

l.66: If available, additional information on the mechanisms of resistance could be provided as this is highly relevant for the choice of compounds to be used in the future. Is resistance acquired e.g. by degradation/modification of the compounds, modification of the target structures of by efficient export?

Figure 4: Stereo information for compound 13 in missing (compared to compound 14).

l.127: Please explain the abbreviation SAR.

l.133: “bacterium”

l.148: “in the above-mentioned period”

l.168: add commas in front of “respectively” (two times). Check entire manuscript (l. 171, etc.).

l.281: “extremophilic"

l.305: avoid “we” as this could be misinterpreted in a way that all compound were identified by the authors (which is not the case).

l.326: “for which” instead of “where”

In the discussion section, the authors should include a short section on the potential sources of the mentioned compounds in higher amounts, e.g. by chemical total synthesis, extraction from the natural producer or engineered production either in the native strains or using heterologous microorganisms as hosts. Here, the advantages and disadvantages of the different strategies should be should be compared in brief (5-6 sentences). The current version of the manuscript also lacks a critical outlook on this highly relevant topic, also in the light of the decreasing industrial interest in producing compounds with antimicrobial activities.

Author Response

We thank the reviewer for the detailed comments, and below give a point-by-point response.

In the entire manuscript: species names should be given as full names (e.g. Candida albicans) when mentioned for the first time and then the abbreviated genus name should be used (C. albicans). This is convention in the microbiology research community and should also be followed in this manuscript.

This has been corrected throughout the manuscript.

Figure 2: For consistency, the sugar moieties should all be presented as chair presentations (as in natamycin) or as stereo projections (as in the other ones).

This is corrected.

line 61: “In summary” sounds like a conclusion, however, this is only the introduction section. If possible, rephrase this sentence to have a good connecting passage to the main section.

This is corrected.

l.66: If available, additional information on the mechanisms of resistance could be provided as this is highly relevant for the choice of compounds to be used in the future. Is resistance acquired e.g. by degradation/modification of the compounds, modification of the target structures of by efficient export?

We have included information on the common resistance mechanisms to azoles, echinocandins and pyrimidine antifungal agents.

Figure 4: Stereo information for compound 13 in missing (compared to compound 14).

This is corrected.

l.127: Please explain the abbreviation SAR.

This is corrected.

l.133: “bacterium”

This is corrected.

l.148: “in the above-mentioned period”

This is corrected.

l.168: add commas in front of “respectively” (two times). Check entire manuscript (l. 171, etc.).

This is corrected.

l.281: “extremophilic"

This is corrected.

l.305: avoid “we” as this could be misinterpreted in a way that all compound were identified by the authors (which is not the case).

This is corrected.

l.326: “for which” instead of “where”

This is corrected.

In the discussion section, the authors should include a short section on the potential sources of the mentioned compounds in higher amounts, e.g. by chemical total synthesis, extraction from the natural producer or engineered production either in the native strains or using heterologous microorganisms as hosts. Here, the advantages and disadvantages of the different strategies should be should be compared in brief (5-6 sentences). The current version of the manuscript also lacks a critical outlook on this highly relevant topic, also in the light of the decreasing industrial interest in producing compounds with antimicrobial activities.

We added a discussion on the potential advantages of chemical synthesis and engineered biosynthesis.

Round 2

Reviewer 1 Report

The manuscript has been improved. I think it is suitable for publication in Pharmaceuticals.